# Effect of Gas Counter Pressure on the Surface Roughness, Morphology, and Tensile Strength between Microcellular and Conventional Injection-Molded PP Parts

**DOI:** 10.3390/polym14061078

**Published:** 2022-03-08

**Authors:** Jianping Ren, Long Lin, Jing Jiang, Qian Li, Shyh-Shin Hwang

**Affiliations:** 1Mould Research Institute, Taizhou Vocational College of Science and Technology, Taizhou 318020, China; renlinbo411@163.com (J.R.); dragon.lin611@163.com (L.L.); 2National International Joint Research Center Micro-Nano Molding Technology, Zhengzhou University, Zhengzhou 450001, China; jiangjing@zzu.edu.cn (J.J.); qianli@zzu.edu.cn (Q.L.); 3Department of Mechanical Engineering, Chien-Hsin University of Science and Technology, Chung-Li 32097, Taiwan

**Keywords:** gas counter pressure, holding time, microcellular injection molding, surface roughness

## Abstract

Microcellular injection-molded parts have surface defect problems. Gas counter pressure (GCP) is one of the methods to reduce surface defects. This study investigated the effect of GCP on the surface roughness, morphology, and tensile strength of foamed and conventional injection-molded polypropylene (PP) products. GCP is generated by filling up the mold cavity with nitrogen during the injection-molding (IM) process. It can delay foaming and affect flow characteristics of microcellular and conventional injection-molding, which cause changes in the tensile strength, flow length, cell morphology, and surface quality of molded parts. The mechanism was investigated through a series of experiments including tuning of GCP and pressure holding duration. Surface roughness of the molded parts decreased with the increase in GCP and pressure holding duration. Compared to microcellular IM, GCP-assisted foaming exhibited much better surface quality and controllable skin layer thickness.

## 1. Introduction

Microcellular injection molding (Mucell^®^) is a popular process that can solve the shrinkage/warpage problem for parts made by injection molding (IM) [1,2]. However, defects like silver streaks, swirl marks, and surface blisters may appear on the parts’ surface [3,4]. Accordingly, microcellular injection-molded parts cannot be used as visible parts. Rough surface is not the only problem. IM also has other negative effects for other applications. For example, Adhikari et al. studied the influence of the surface roughness of conducting polypyrrole thin-foil electrodes on the electrocatalytic decrease of nitrobenzene [5]. The results showed that roughness of the foils played a significant role in determining their catalytic function. Several efforts have been made to solve this problem. One of them is gas counter pressure (GCP), which was introduced by Balevski et al. in 1978 [6]. Later, Allied Chemical Corporation introduced the Allied process [7]. In GCP IM the gas exists in the cavity before and throughout the whole process, which can avoid polymer melt bubbling and serve as packing pressure. Kim et al. used the GCP technique to get rid of the drying process on conventional IM [8]. The purpose is the restriction of water from vaporizing to gas pockets inside the cavity. In order to achieve that, the GCP must be larger than the certain compression of water until the polymer is consolidated. Bledzki et al. [9] utilized the GCP in microcellular IM, which reduced the product’s external roughness from 23 to around 1 μm. The weight was also reduced from 12.8 to 10.2 wt%; this also helped to stabilize the cell size. In 2004, Ohshima used supercritical CO_2_ as a GCP source and the CO_2_ fluid could dissolve into the molten polymer [10]. Therefore, the molten polymer’s properties were changed. Bledzki et al. [11] used unusual processing techniques such as MuCell^®^, GCP, and precision core back (PCB) to study the impact behavior of polycarbonate. The results show that the cell size in MuCell^®^, MuCell^®^+GCP, and MuCell^®^+GCP+PCB processes are 47–85, 23–68, and 1.9–10 μm, respectively. Except for the pressure method, in mold decoration [12,13] and co-injection molding techniques [14], which lead to unfoamed skin with a foamed core, there are other options to improve the surface quality.

The nucleation theory of microcellular foaming was proposed by Colton et al. [15].

They adopted Gibbs free energy theory and constructed two models for homogeneous and heterogeneous nucleation. The homogeneous nucleation is described as follows:(1)$$△G_{hom}^{*}=\frac{16\pi }{3{\left(p_{g\;}-p_{t}\right)}^{2}}\gamma_{bp}^{3}$$
where $△G_{hom}^{*}$: critical free energy barrier (*1/s*); *γ_bp_*: surface energy of the polymer-bubble interface (*J/cm^2^*); *p_g_*: interior gas pressure inside the bubble (*bar*); and *p_t_*: melt pressure outside of the bubble (*bar*).
(2)$$N_{hom}=f_{0}C_{0}\exp\left(-△G_{hom}^{*}\right)/kT$$
where *N_hom_*: homogeneous nucleation rate (*nuclei/(m^−3^s*)); *f*_0_: frequency factor for the homogeneous nucleation; *C*_0_: concentration of gas molecules in solution (*cluster/m^3^*); *K:* Boltzmann’s constant (*J/K*); and *T:* the absolute temperature (*K*). 

Polymer foams can be achieved either by physical blow agent (PBA) or chemical blow agent (CBA). However, equipment for PBA is more costly than for CBA. Llewelyn et al. [16] used PBA and CBA methods to study the surface roughness improvement by processing parameters of injection molding. Melt temperature was found to be the most effective parameter to reduce the surface roughness.

The principle of gas counter pressure (GCP) is shown in Figure 1. In general, in microcellular injection-molding, when the melt is being forced out of the barrel, the melt front takes the form of a fountain flow. The micro-bubbles in the middle part are pushed forward along with the flow, but the flow near the mold wall is slower due to lower temperature. Therefore, the middle melt is overturned towards the mold wall, and some bubbles run to the product surface. However, if there is GCP in front of the melt, foaming behavior will be very different. When the counter pressure is very low, the melt foams freely in the filling stage. Especially, when the pressure is lower than 1 atm, there will be silver streaks on the product’s surface [17]. The melt front changes from bullet-shaped into a slightly convex shape. Free foaming will be constrained if GCP is between 1 bar and the critical pressure, which leads to single phase melt. It does not foam if the GCP is greater than the critical pressure. A recent study [18] showed when the GCP is higher than a constant, the product surface may not foam. Besides that, the gas holding time also plays an important role in restricting the foaming process. In terms of the other uses of counter pressure, the amount of super-critical fluid dissolved by the melt is critical, which can be measured under counter pressure [19]. 

In addition to the GCP method, high mold temperature is another option to improve the external roughness of the shot-formed products. However, this will result in a longer cycle time and is not preferred for mass production. Turng et. al. used polytetrafluoroethylene (PTFE) film as the insulator and kept the cavity temperature high. The parts’ surface quality was improved significantly [20]. Cha and Yoon used several mold temperatures to eliminate surface swirl marks on the microcellular parts [21]. Dynamic mold temperature control (DMTC) was applied to the foam molding to improve the surface as well, where the mold temperature is high and assisted cooling is also fast. In our previous work [22], both the GCP and DMTC processes were applied on polystyrene. The part’s surface quality was improved and frozen layer thickness was reduced as well. Park et al. [23] used the pressure method to study the foaming mechanism of microcellular foams with GCP and mold-opening processes, which can produce a high-quality surface.

Microcellular injection technology [24,25,26] is a special process to overcome some problems that the conventional injection molding cannot. Most of the microcellular foam plus GCP technology [27,28] emphasizes improving surface roughness problems. Very few papers discuss the effect of GCP on the flow length, tensile property, and gas penetration problem. In this study, the surface quality of microcellular foam is controlled using a pressure–temperature path control method. Prior studies only discussed the effects of temperature [27] or pressure [16]. This study investigates combined effects by temperature and pressure, aiming at optimizing parts’ surface quality in the foaming process. Therefore, the GCP and mold temperature control system was built. This system comprises four units: (1) microbubble unit; (2) sealed core unit and a device allowing the counter pressure to enter and leave; (3) process signal collecting system; and (4) mold temperature control system driven by water. The effect of GCP on the tensile, morphology, and surface roughness properties of foamed/conventional injection-molded PP parts was systematically investigated. It turned out that surface quality was well tuned. Furthermore, tensile property and flow length were comprehensively discussed, which is rarely reported based on previous work regarding PS molding [22].

## 2. Experimental

An Arburg 420C, 100-ton molding machine (Lossburg, Germany) installed with a microcellular injection molding capability was utilized to manufacture the samples. A GCP control unit (Polypro Technology Inc., New Taipei, Taiwan) equipped with an air control gate bound to the cavity was adopted to regulate then control the on-line air pressure within the mold. The cavity needs an o-ring, which is to keep the cavity sealed. The melt gate is located on the top of the cavity and the gas inlet is on the bottom side (Figure 2).

### 2.1. Materials

Semi-crystalline polymer, PP, K1035 with MFI of 39 g/10 min, was obtained from Taiwan Plastics Corp. Ltd., Taiwan. This PP has a melt temperature of 210 °C and density of 900 kg/m^3^.

### 2.2. Cavity for Gas Counter Pressure 

The tensile bar dimension according to ASTM D 638-98 is 100 × 30 × 3.2 mm (L × W × T). Figure 2 shows the tensile bar cavity configuration. The gate is situated on the top side. On the lower side of the specimen the gas is input. To acquire the instantaneous temperature and compression in the mold throughout the foaming with GCP practice, a temperature sensor is located near the top bell end, which is close to the gate position, and the pressure sensor is located near the lower bell end. There is an overflow area between the lower bell end of the tensile bar and gas inlet. This is to make sure that the tensile bar has enough length for the tensile test. A temperature-resistant o-ring was adopted to avoid gas leaks from the mold cavity (Figure 2). 

### 2.3. Gas Counter Pressure Device

The GCP device includes a gas booster (model AG62) by Haskel International Inc. (Burbank, CA, USA), gas tank, and controller. The booster pump can boost the nitrogen gas up to 32 MPa. The controller can adjust the pressure range and holding time. Injection was delayed for 5 s, during which the mold cavity would be filled with counter gas.

PP with a melt flow index of 39 was used in this investigation and molded by traditional and microcellular molding processes. The foaming process was operated by a 100-ton German ARBURG injecting molding machine, which had a function of microcellular foam molding ability. Nitrogen was adopted as the foaming agent and counter pressure gas. Table 1 shows process conditions for traditional, microcellular, and GCP moldings. The flow rate of SCF nitrogen was 0.131 kg/h and the weight reduction rate was 10 wt% for this study.

### 2.4. Instrumentation

ASTM D 638-98 was used as the tensile test standard of the mechanical property and the tests were executed on HT-9102M of a Hona-da universal test machine (Taichung, Taiwan). Tensile test data were the average of ten tests. The specimens’ tensile values closest to the average value were chosen as the candidates for cell morphology and surface roughness inspections. The cell morphology of the foamed specimens was investigated by a Hitachi S-3000N scanning electron microscope (SEM) (Tokyo, Japan). The sample was sliced into tiny fragments and plated with Aurum before being investigated with SEM. Surface roughness was scanned by a Keyence VK-8510 (Osaka, Japan) laser 3D profile microscope.

## 3. Results and Discussion

Our investigation tried to correlate the interactions between processing, morphology, and properties of PP and the foamed specimens manufactured in MuCell^®^ and traditional injection-molding processes with/without GCP. The GCP has a certain effect on the melt advance during the IM process. The mold pressure must be acquired before the experiment. Figure 3 shows the mold pressure traces under different holding times for a GCP of 100 MPa. It can be seen that there is a time delay of injection for 5 s and a pressure surge after injection. The pressure surge indicates that the melt advancing through the pressure sensor and cavity pressure is in the range of 100 MPa. There is no packing pressure during the microcellular injection-molding process with GCP.

### 3.1. Microcellular Foaming

There is a constraint on the cell growth if there is a GCP inside the cavity upon microcellular injection. Figure 4 shows normal SEM micrographs of the rupture surfaces vertical to the flow orientation of the foamed PP parts with the magnification of 25 (left) and 75 (right). Figure 4A1 (left) is the part without GCP while Figure 4B1 is the part with GCP and a holding time of 0 s. Figure 4C1,D1 are the parts with holding times of 10 and 20 s, respectively. There are several large cells in the core section of the part without GCP. The part without GCP has the thickest skin layer among the four Figures. The skin layer thickness decreases as the holding time increases. This is caused by gas penetration into the melt during the melt filling process [23]. However, Li et al. [29] observed several pits caused by cell rupture and many gas marks on the melt front when not using GCP, although they used CBA foaming. There is a series of cells along the transition area between the skin and core layers of the part with a holding time of 10 s as shown in Figure 4C1. This is the cell compressed by the GCP around that area. Once the pressure is released, the unfrozen and compressed cells start to grow again. However, this phenomenon is not obvious on the part with a holding time of 20 s. 

Figure 4A2 (right) is the part without GCP and Figure 4B2 is the part with GCP and a holding time of 0 s. The cell size decreases and cell density increases considerably when the holding time increases from 0 to 10 and 20 s as shown in Figure 4C2,D2. The average cell size for parts with holding times of 0, 10, and 20 s are 74.83, 17.63, and 13.08 μm, respectively. The cell size distribution of Figure 4 is shown in Figure 5. Shiu et al. [30] used simulation software, Moldex3D, to predict the cell size distribution on microcellular foams with GCP and DMTC processes. The cell size distribution is in agreement with the experiments. The parts having a small cell size will have better tensile strength [24] as shown in Figure 5. Table 2 shows the relationship between cell size and skin layer thickness microcellular injection molding with GCP of 100 bar and different holding times. As the holding time increases, the cell size and skin layer thickness decrease from 74.83 to 13.08 μm and 0.55 to 0.45 μm, respectively, with the holding time increment from 0 to 20 s. However, the skin layer thickness decrement agrees with results from Rodrigue et al. and Chang [31,32] and is different to the results from Hou et el. with the mold-opening process [33].

### 3.2. Surface Roughness 

Surface roughness is an important factor in computer, communication, and consumer electronics manufacturing, especially if the part is used as the visible part. The site of foamed parts that was examined by surface roughness and a representative lateral view of the foamed structure are shown in Figure 6a,b,separately. Figure 7 shows the surface roughness measurement at three different positions: near gate, mid, and flow end position, both for solid and microcellular injection-molding without GCP. The surface roughness at the gate, mid, and flow end are 2.52, 2.71, and 4.84 μm, respectively, for solid molding. The gate has the best surface roughness among the three positions. The reason is that the temperature at the gate is higher than that of the flow end, which results in a smoother surface [34]. The surface roughness at the gate, mid, and flow end are 49.4, 47.37, and 45.69 μm, respectively, for microcellular injection-molding. The microcellular injection-molding has silver streaks, swirl marks, and blisters on the surface, which are caused by microcells on the melt front of the core section trapped by the mold wall (this is the fountain flow effect in the filling process), thus leading to the rough surface [29,35]. The gate has the highest surface roughness of the three positions. It has different trends of surface roughness compared to that of solid molding. This is due to the temperature at the gate being higher than that at the flow end and the cell trying to escape to the surface at the gate, so the skin layer thickness at the flow end is thicker than that at the gate for microcellular molding. This causes the surface roughness at the gate to be higher than that at the flow end. However, the solid molding has better overall surface roughness than that of microcellular molding. A laser 3D profile microscope is expensive and needs careful operation, so a surface gloss instrument is used to depict the surface roughness value. Li et al. [29] used chemical foaming, gas counter pressure, and surface gloss methods to study the surface roughness of foamed parts. Their results show that surface gloss values are 6.2 and 11.8 for counter pressure of 0 and 0.6 MPa, respectively. Chen et al. [36] used a laser 3D profile microscope and surface gloss instruments to measure microcellular injection-molded high-density polyethylene/wheat straw composites. The results from both methods were in agreement. Their surface gloss value were 15.9, 20.3, 26.5, and 29.4 GU for holding times of 0, 0.5, 3.5, and 6.5 s, respectively.

Table 3 shows the surface roughness comparison of different GCP and holding time at the middle position for microcellular molding. It is seen that as the GCP value increases, so does the surface roughness at the same holding time. If the GCP increases from 0 to 10 MPa, the surface roughness decreases from 15.11 to 11.92 μm at the holding time of 10 s, so the improvement is 21%. Figure 8 exhibits the visual surface roughness comparison of the sample under different GCPs at the holding time of 10 s. It seems that the silver streaks can be eliminated, which is caused by no cell rupture on the melt front through the filling process when the GCP is higher than 5 MPa. This value is very close to the value of 45 bar reported by Li et al. [37].

Table 4 shows the surface roughness measurement at the gate, mid, and flow end positions both for solid/microcellular molding with GCP and different holding times. It shows a surface roughness decrease with an increase in holding time. If the holding time increases, the skin layer thickness decreases as well, facilitating better surface roughness. Figure 9 demonstrates the skin layer depth measurement at the middle position of the microcellular molding with/without GCP and different holding times. The skin layer thickness without GCP is 0.72 μm and it decreases to 0.58, 0.52, and 0.48 μm with GCP and holding times of 0, 10, and 20 s, respectively. The results show that the sample with GCP has a smaller skin layer thickness than that of the ones without GCP, and there is a continuous decrement with increasing holding time. This is caused by the penetration of compressed gas into the melt front during the filling process. Thus, it serves as the heat insulation of the gas between the polymer melt and cavity wall [32]. If the holding time increased, more compressed gas would enter the melt resulting in decreases in the skin layer thickness.

### 3.3. Flow Length Comparison

For the conventional (solid) injection molding, the melt advances without any obstacles during filling, so the melt front has the fountain flow characteristics as shown in Figure 10. GCP has a certain effect on the flow length of the injection molding because of the pressure on the melt front. Figure 10 shows the flow length comparison of the microcellular/solid injection molding with and without GCP at the same shot size. At the top, two tensile bars are injection-molded without GCP; the first one is molded by solid molding while the second one is molded by microcellular molding. The flow length of the solid molding is less than that of the microcellular molding. This is caused by the cell growth during foaming, which makes the melt flow longer than that of solid molding. When GCP is introduced into the cavity, it has different trends as shown in Figure 10. The third and fourth ones are solid and microcellular molding, respectively, with GCP and a holding time of 0 s. The flow length of the solid is longer than that of the microcellular one. The driving force of the cell growth during foaming is not strong enough to resist the GCP, which makes the flow length shorter on the foaming sample, and there are some bubbles on the flow end where gas penetrates into the melt front on the solid sample. This phenomenon is more severe if the holding time increases, which is shown in Figure 10. However, there is an interesting phenomenon for the holding time of 20 s. The flow length of the 20 s samples is longer than that of the 10 s samples. This can be explained as there is an internal resistance force. Once the GCP is released, the melt bounces back and makes the flow length longer. Figure 11 shows the flow length comparison of the previous figure. Up to now, there are no reports about the effect of GCP on the flow length comparison.

### 3.4. Tensile Test

The tensile strength of the polymer depends on the molecular chain orientation [38]. If the molecular chain orientation is more aligned, the tensile strength is higher. However, if there is a GCP against the melt front, the molecular chain orientation will be distorted [39]. This will have a significant effect on the tensile strength and make the material strength decrease. However, there are few papers about GCP effects on tensile strength. Figure 12 shows the stress–strain curve of microcellular injection-molded PP with a counter pressure of 50 bar and different holding times. The microcellular part without GCP has the largest tensile strength of 26.33 MPa. The tensile strength decreases to the lowest value of 19.04 MPa with GCP and a holding time of 0 s. Then tensile strength increases to 22.53, 22.81, and 23.81 MPa accordingly with increasing the holding time to 5, 10, and 20 s, respectively. The GCP in front of the melt front is the main factor that distorts the molecular orientation and in turn makes the strength decrease [38,40]. Although the tensile strength increases, the elongation of material increases from 10.88 to 13.98 % with the increase in holding time from 0 to 20 s. Therefore, the increase in holding time leads to enhanced mechanical properties when the material is injection-molded with GCP. The GCP also serves as the packing pressure as the conventional injection molding does. However, packing pressure takes place after injection molding to prevent the shrinkage before cooling. As the packing time increases, so does the tensile strength at the same packing pressure [40]. 

Figure 13 shows the results with GCP of 100 bar. It has a similar trend that an increase in holding time made the material stronger as shown in Figure 12. However, the longest elongation is 17.2 mm at the holding time of 5 s instead of 20 s. Figure 14 shows the tensile test results with GCP of 150 bar. Those curves have a trend similar to the tensile test with GCP of 100 bar, i.e., the longest elongation takes place at a holding time of 5 s and the tensile strength increases with an increase in holding time. The tensile strength of GCP-assisted samples with a holding of time 20 s is 24.7 MPa larger than that (17.26 MPa) of the part without GCP. Accordingly, the packing material dense effect (increase in holding time) is higher than the molecular orientation effect in turn making the strength larger. Figure 15 shows the tensile strength comparison of the microcellular foamed PP with/without GCP and different holding times. It is a tensile strength value combination of Figure 10, Figure 11 and Figure 12. The molding with GCP of 150 bar has the largest tensile strength among 3 different GCPs. For the high GCP molding, it needs a large shot size otherwise the specimen is short shot. The shot sizes for molding with GCP of 50, 100 and 150 bar are 14.5, 16, and 18 cm^3^, respectively. The tensile strength is also dependent on the shot size. The larger the shot size, the higher the tensile strength. The GCP and holding time values are the key factors to determining the tensile strength as the packing pressure and packing time function in conventional injection molding. All the detailed tensile strength data are listed in Table 5 and Figure 15.

Figure 16 exhibits the variation in weight reduction ratio (WRR) at various GCP and holding times. WRR is also related to the cell morphology and tensile strength properties. The trend shows that as the holding time increases, the weight reduction decreases. At the beginning of the experiment the WRR is set as 10 wt% and weight is 7.25 g without GCP. The weight and WRR are 5.53 (31.37%), 5.13 (36.25%), and 5.59 g (30.64%) for 5, 10, and 15 MPa, respectively, of GCP at the packing time of 0. There was compressed gas penetrating into the sample, otherwise the weight would not be less than 7.25 g. As the packing time increases, it needs more shot size to prevent small size and causes the decrease in WRR. However, low WRR has higher tensile strength by comparing Figure 12, Figure 13 and Figure 14. One can obtain high WRR by mold-opening microcellular injection-molding [33].

## 4. Conclusions

In this study, the properties of GCP and holding time on the tensile strength, morphology, and surface roughness of injection-molded PP foams were investigated. We found that GCP could serve as a packing pressure during the filling process as conventional injection-molding does. GCP can affect the flow length of the melt and gas can penetrate into the melt through the melt front during the filling process. Accordingly, as GCP or holding time increase, so does the tensile strength. However, the tensile strength of the foamed part without GCP is larger than the ones with GCP due to more molecular chain orientation. The increase in holding time and GCP led to the reduction in the cell size and surface roughness. With proper GCP and holding time, surface roughness of foamed PP would be much closer to the ones molded using IM, which exhibit great potential in industrial manufacture.

## Figures and Tables

**Figure 1 polymers-14-01078-f001:**
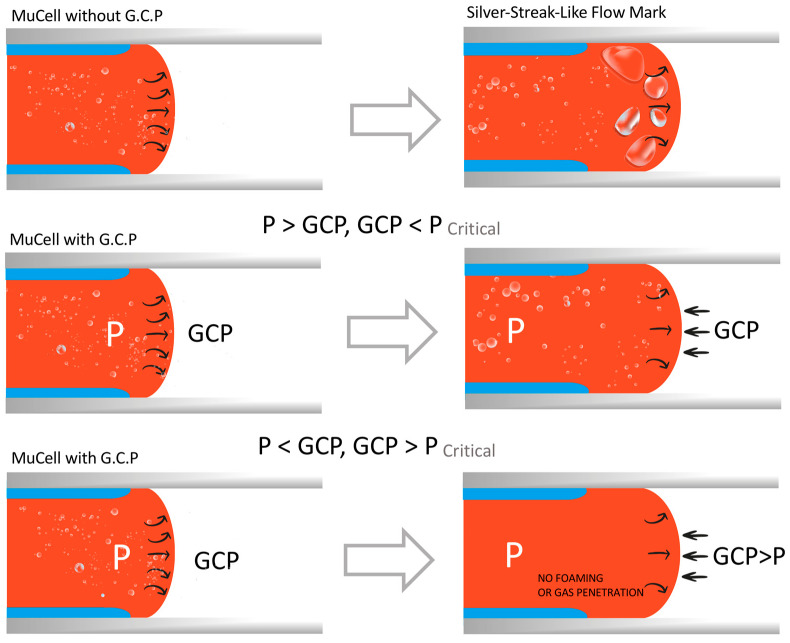
Effects of GCP control on the foam structures.

**Figure 2 polymers-14-01078-f002:**
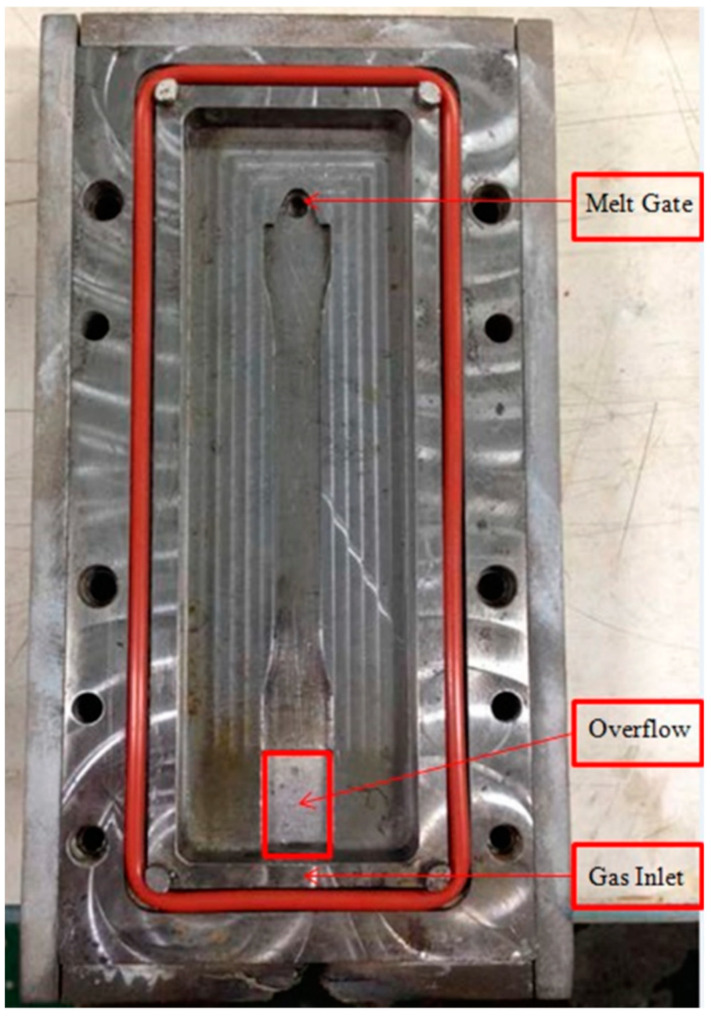
Cavity for gas counter pressure experiment.

**Figure 3 polymers-14-01078-f003:**
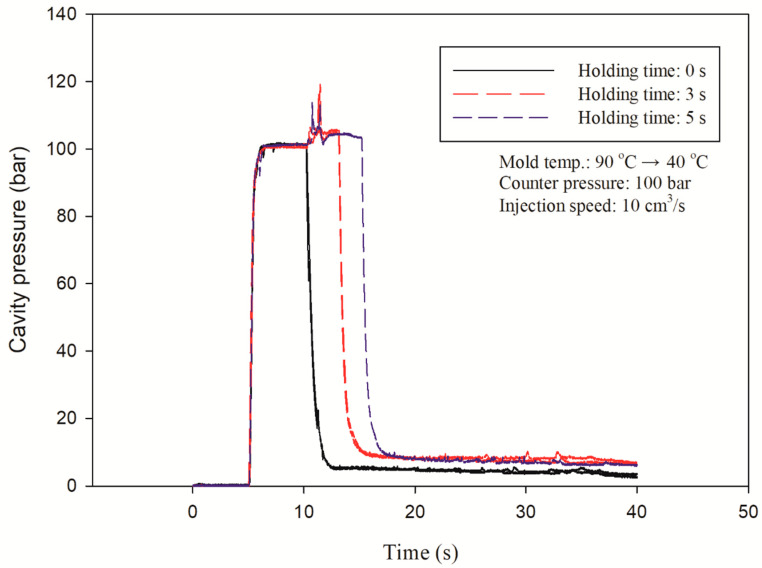
Cavity pressure traces under different holding times for GCP of 100 bar.

**Figure 4 polymers-14-01078-f004:**
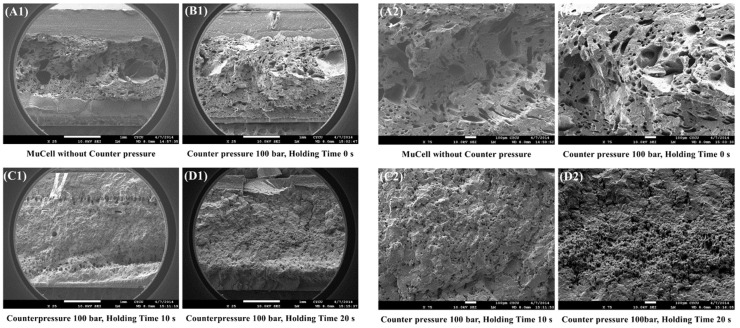
SEM pictures of fracture surface perpendicular to the flow direction with different holding times at GCP of 100 bar. Left (×25), right (×75).

**Figure 5 polymers-14-01078-f005:**
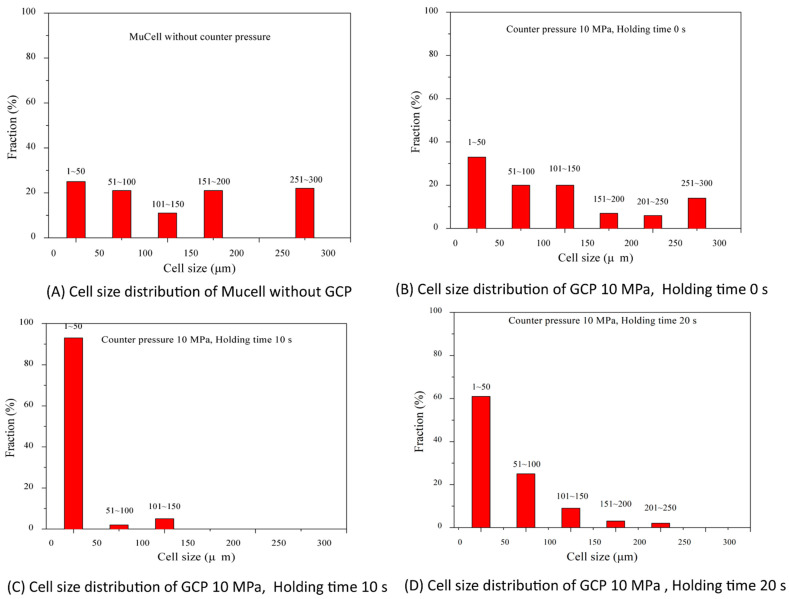
Cell size distribution of Figure 4.

**Figure 6 polymers-14-01078-f006:**
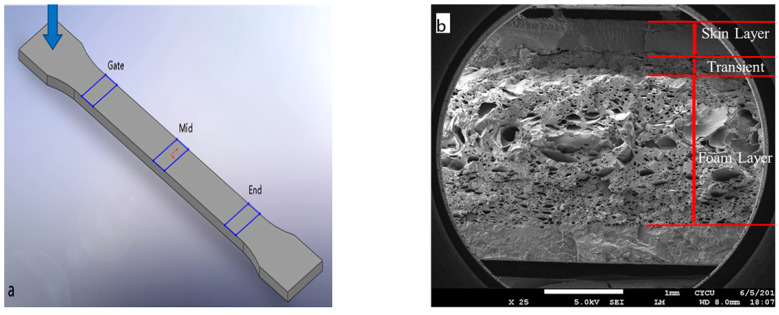
(**a**): Examination locations for parts’ surface roughness; (**b**): a typical SEM micrograph of a foamed sample.

**Figure 7 polymers-14-01078-f007:**
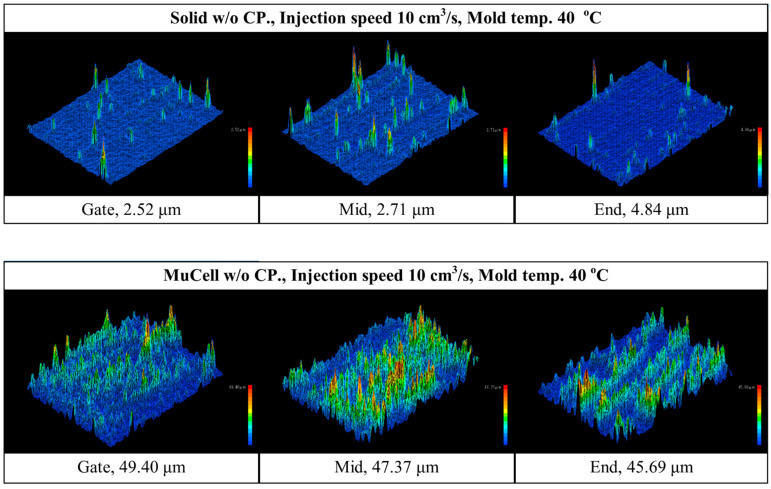
Surface roughness measurement at gate, mid, and flow end positions both for solid and microcellular molding without GCP.

**Figure 8 polymers-14-01078-f008:**
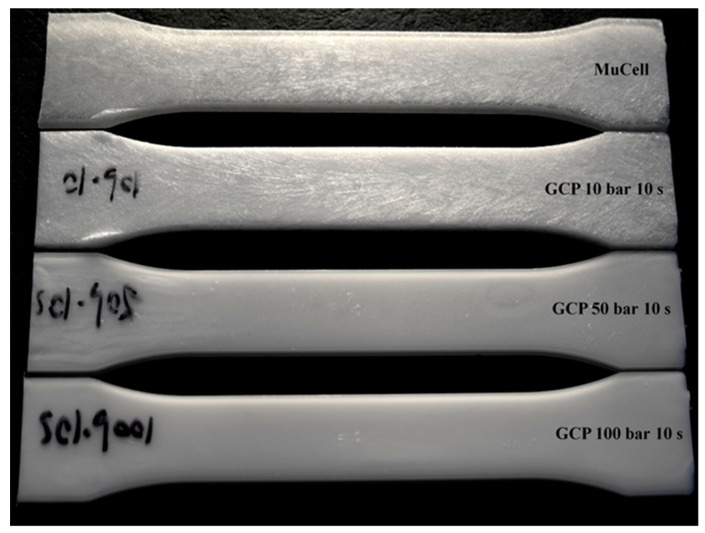
Surface visual quality of the microcellular molded parts at different GCPs.

**Figure 9 polymers-14-01078-f009:**
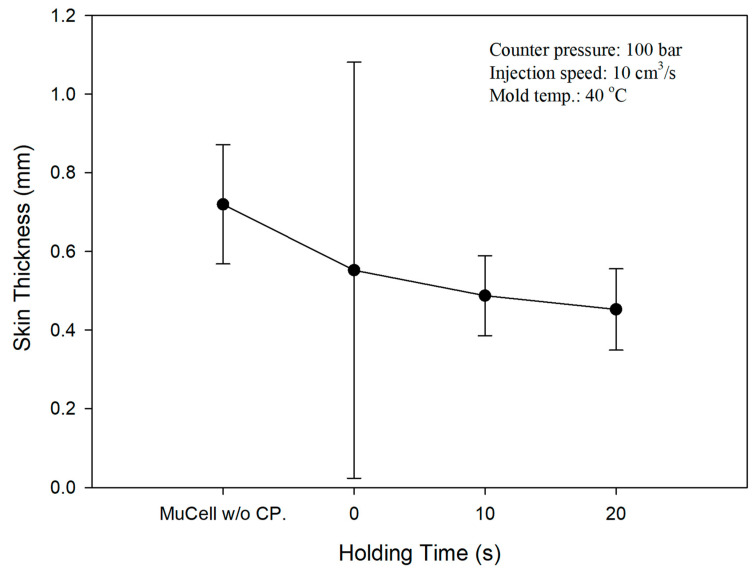
Skin layer thickness measured at the middle position of the microcellular molding with/without GCP and different holding times.

**Figure 10 polymers-14-01078-f010:**
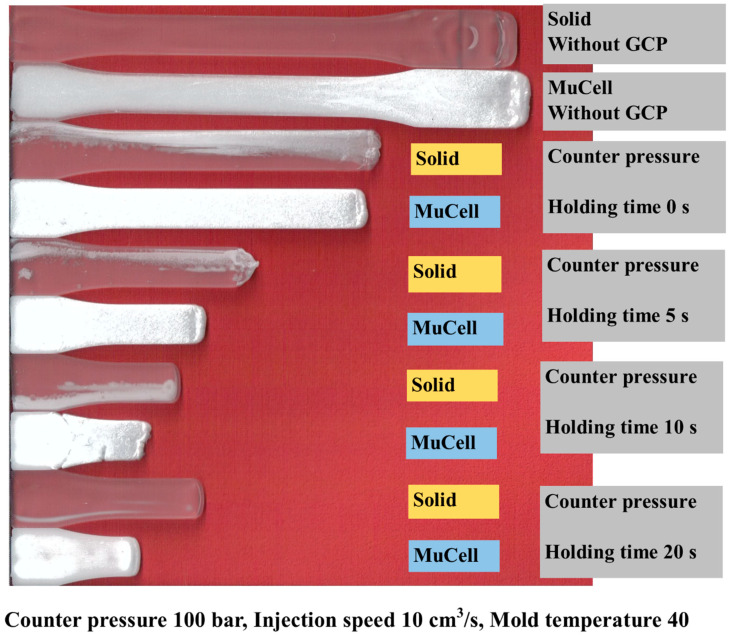
Visual flow length comparison for solid/foam molding with/without GCP.

**Figure 11 polymers-14-01078-f011:**
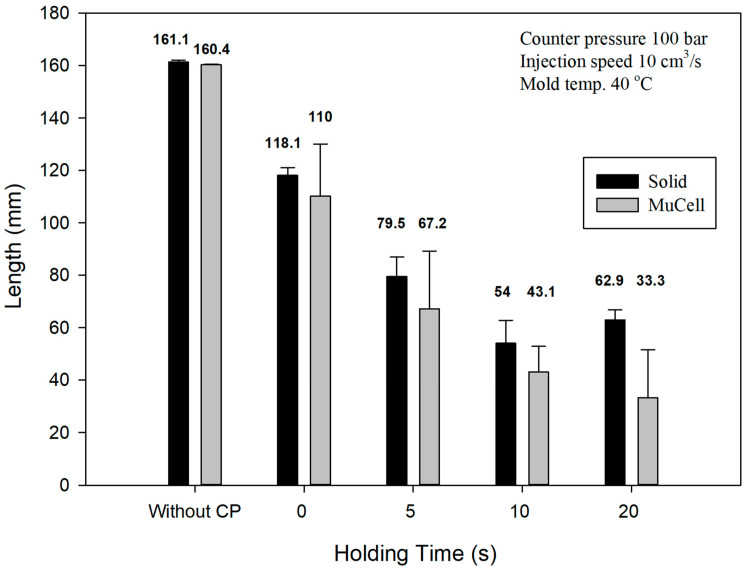
Flow length comparison for solid/foam molding with/without GCP and different holding times.

**Figure 12 polymers-14-01078-f012:**
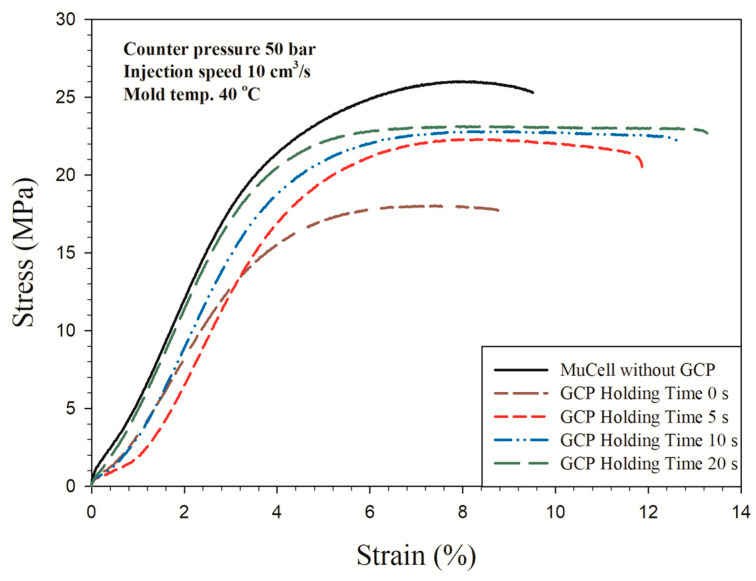
Stress–strain curves of microcellular injection-molded PP with GCP of 50 bar and different holding times.

**Figure 13 polymers-14-01078-f013:**
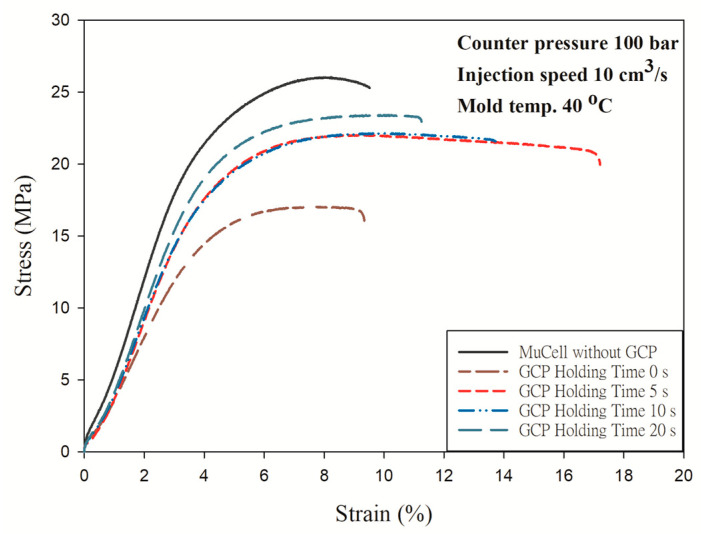
Stress–strain curves of microcellular injection-molded PP with GCP of 100 bar and different holding times.

**Figure 14 polymers-14-01078-f014:**
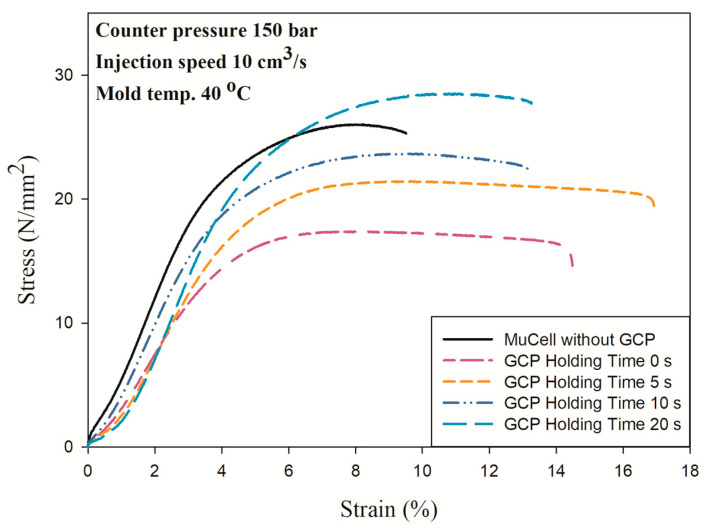
Stress–strain curves of microcellular injection-molded PP with GCP of 150 bar and different holding times.

**Figure 15 polymers-14-01078-f015:**
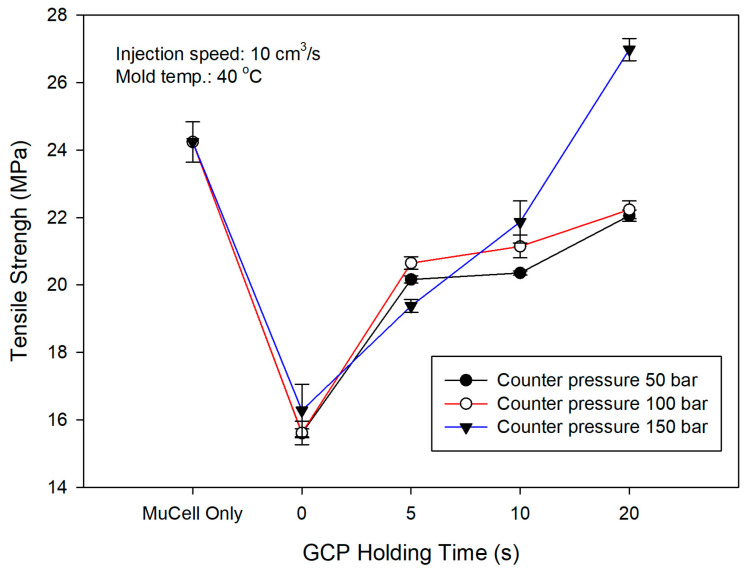
Tensile strength comparison of microcellular injection-molded PP with different GCPs and holding times.

**Figure 16 polymers-14-01078-f016:**
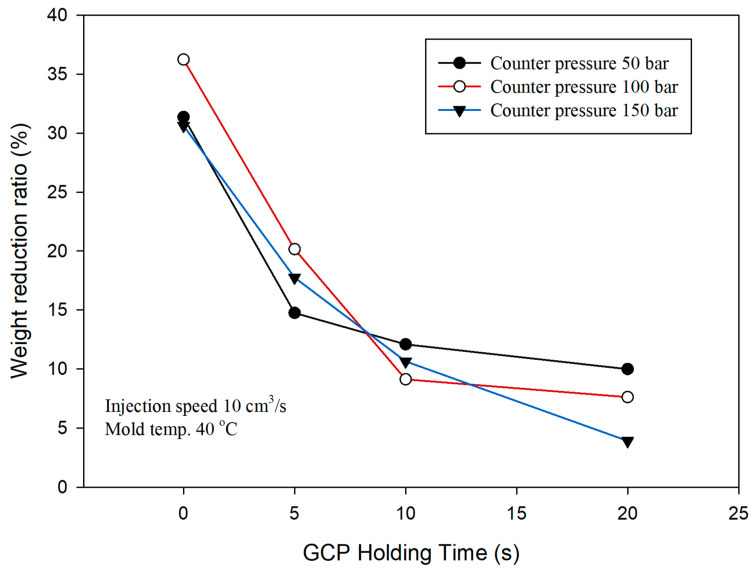
Comparison of weight reduction ratio under various GCP and holding times.

**Table 1 polymers-14-01078-t001:** Process parameters both for conventional and Mucell molding processes.

Process Parameters	Solid	Foamed
Shot size (cm^3^)	20	14
Melt temperature (°C)	210	210
Mold temperature (°C)	40	40
Injection speed (cm^3^/s)	10	10
Injection pressure (bar)	1000	1000
Packing pressure (bar)	500	–
Back pressure (bar)	50	185~188
Screw speed (m/min)	15	15
Cooling time (s)	25	30~40
SCF content (wt%)	–	1
SCF flow rate (kg/h)	–	0.131
Weight reduction ratio		10%
Injection delay time (s)	5	5
GCP (bar)	50/100/150	50/100/150
GCP holding time (s)	0/5/10/20	0/5/10/20

**Table 2 polymers-14-01078-t002:** Average cell size and skin thickness of the microcellular injection molding with GCP and different holding times.

GCP (Bar)	Holding Time (s)	Cell Size (μm)	Skin Thickness (μm)
100	0	74.83	0.55
	10	17.63	0.49
	20	13.08	0.45

**Table 3 polymers-14-01078-t003:** Surface roughness measurements with different GCP and holding times on the middle position.

GCP (bar)	GCP Holding Time (s)/Ra (μm)			
	0	5	10	20
0	37.69	17.68	15.11	8.90
10	37.00	15.04	12.96	7.66
50	35.68	14.50	12.39	7.33
100	34.55	14.00	11.92	7.02

**Table 4 polymers-14-01078-t004:** Surface roughness measurements at gate, mid, and flow end both for solid/microcellular molding with GCP and different holding times. Unit: μm.

	Process	Solid	MuCell W/O GCP	Mucell (GCP 100 bar)/Ra (μm)			
Position				0 s	5 s	10 s	20 s
Gate		2.52	49.4	37.02	15.49	12.11	9.08
Mid		2.71	47.37	34.55	14.00	11.92	7.06
End		4.84	45.69	30.78	12.37	9.88	7.04
Average Surface Roughness		3.36	47.49	34.12	13.95	11.30	7.73

**Table 5 polymers-14-01078-t005:** Tensile property results of Figure 12, Figure 13 and Figure 14.

	Holding Time (s)	Tensile Strength (MPa)	Elongation (%)	Young’s Modulus (MPa)	STD
Mucell without GCP		24.33	12.77	21.78	0.6
GCP 50 bar	0	19.14	10.88	15.07	0.13
	5	22.53	16.92	18.67	0.10
	10	22.81	18.01	18.88	0.06
	20	23.81	13.98	21.50	0.17
GCP 100 bar	0	17.26	9.63	16.21	0.46
	5	21.76	17.2	20.06	0.32
	10	22.73	13.9	21.88	0.88
	20	24.70	11.91	23.83	2.17
GCP 150 bar	0	17.26	21.3	14.99	0.78
	5	19.37	20.77	17.19	0.19
	10	21.86	20.16	20.01	0.62
	20	27.50	15.79	22.28	0.33

## Data Availability

The data presented in this study are available on request from the corresponding author.

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
