# Peer review of "Effect of Gas Counter Pressure on the Surface Roughness, Morphology, and Tensile Strength between Microcellular and Conventional Injection-Molded PP Parts"

_polymers, 2022, doi:10.3390/polym14061078_

Round 1

Reviewer 1 Report

The article “Effect of gas counter pressure on the surface roughness, morphology, and tensile strength between microcelluar and conventional injection molded PP parts” is clearly written and can be accepted for publication after minor corrections.

The  article deals with the processing of polymers and can provide valuable information for engineers and scientists working in this industry.

Specific comments:

  1. Why the melt temperature (Table 1, 210 oC) was lower than the PP melting temperature (2.1. Materials. PP has a melt temperature of 230 ℃).
  2. The aspect of molecular orientation requires the support of relevant literature.

Author Response

response file of the comments is attached.

Reviewer 2 Report

Dear authors,

Congratulations for this good paper

I have two comments:

1) The distribution of the cell size is certainly also important. You write that these values were formed from an average. But I can't find the confidence interval and the number of measurements. Please add that
2) The same applies to mechanical measurements. Please add the number of specimens here.

thank you
The reviewer

Author Response

Response file is attached. 

Reviewer 3 Report

The authors nicely structured the paper and described a comparison of how gas counter pressure affects some properties. But the work has some small things that need to be fixed, and unfortunately a little bigger. Which I believe every reviewer will emphasize.

My comments:

  1. The first sentence in Abstract should be reformulated (line 11). A sentence cannot start with An ASTM tensile bar was chosen……, it should start with the point of the paper, and if it starts with ASTM, then the number of that standard should be written.
  2. Sort the keywords alphabetically. See the instructions of journal if each keyword is capitalized.
  3. The space between two words is missing in a few places (lines 17, 65, 93, 106, 289). But also in a picture 2 (counterpressure).
  4. Is the term silver streaks or sliver streaks? There are a lot of places in the work of the word sliver.
  5. Why is the font in lines 48 and 183 different from the rest of the text?
  6. When equations are given, the symbol, name and unit of measure should be written below. No units of measurement are mentioned anywhere, eg p - pressure (Pa, bar). And all symbols should be italic (line 53-55).
  7. Before Chapter 2, a short chapter with a few sentences should be written to introduce the reader to what the authors have done differently and new than what was written in the Introduction.
  8. Line 109: when the dimensions of the test specimen are mentioned, it should be written immediately and according to which standard it was made. Here the number of the standard is a one page behind that. The authors also mention (line 110: …….common depth for microcellular injection molding. Does this mean that this depth was not made according to the standard? If that, this should be written. Why word common?
  9. Line 111: gate is situated on the top side. - It would be good to insert a picture with a view of that.
  10. Line 124: PP with melt flow index of 39? What is the unit of measurement? 39 - what?
  11. Line 129: kilogram is written in lower case kg, not Kg.
  12. when stating the number of the ASTM 638 standard, the year of the standard should also be stated. Authors should check whether the actual standard name is ASTM STD-638 or ASTM D638.
  13. Line 135: the authors list 10 test specimens. Why do 10 test specimens apply? Isn't it enough according to the standard 5.
  14. Line 169: isn't that conclusion logical?
  15. Line 180: what does 3C mean?
  16. Line 208-209: talk about holding time from 0 to 10 and then write the units of measurement MPa. The authors need to verify this statement.
  17. When it comes to surface roughness, is it measured Ra or Rz (Ra - average roughness of a surface or Rz - difference between the tallest (max) peak and the deepest (min) valley in the surface)?
  18. If everything is measured on 10 test specimens for each design test then should the roughness not have a mean value and a standard deviation? That is, in the paper it is obligatory to insert a table with the values ​​of all 10 measurements - mean value + standard deviation. Especially for tensile properties (Chapter 3.4).
  19. Why is Figure 8 written in red in the text?
  20. In Figure 9 - values should be written on the value bars.
  21. The title of the Figure 9 itself should be placed on the previous page next to the figure.
  22. When examining tensile properties, did a fracture occur? If it is why there is no results on the stress on break and modulus? Put a table with values ​​+ standard deviation. Put the results for 50, 100 and 150 bar in the same table to make it easier to follow the results.
  23. The legends of Figures 10, 11 and 12, or rather the line, should be the same colour in my opinion. For example: Mucel without GCP pink in all pictures, etc.
  24. Line 314-315: WRR is also related to the cell morphology and mechanical properties (tensile and impact strength) - impact strength? - it is necessary to substantiate this with some literature. Because the authors themselves did not research it.
  25. The biggest complaint is that the Discussion chapter is missing. Refer to previous works and what they did. This raises the paper to the scientific level required for the journal Polymers.
  26. Also as a remark for the references are mostly very old works. Don’t get me wrong and I also think they are helpful. But that no one has been involved in the Mucell procedure for about 10 years? Some recent references should be found and certainly used in the Discussion section.
  27. In the references list: the titles of papers from the journal are missing.

Author Response

Response file is attached.

Round 2

Reviewer 3 Report

The authors have changed a lot of things from the first submission. 

The authors answered to all my questions.

However, the experimental part, discussion and conclusion had to be separated into separate chapters.